# *Pseudomonas aeruginosa* Toxin ExoU as a Therapeutic Target in the Treatment of Bacterial Infections

**DOI:** 10.3390/microorganisms7120707

**Published:** 2019-12-16

**Authors:** Daniel M. Foulkes, Keri McLean, Atikah S. Haneef, David G. Fernig, Craig Winstanley, Neil Berry, Stephen B. Kaye

**Affiliations:** 1Department of Eye and Vision Science, Institute of Ageing and Chronic Disease, University of Liverpool, Liverpool L7 8TX, UK; atikahh@liverpool.ac.uk (A.S.H.); sbkaye@liverpool.ac.uk (S.B.K.); 2Department of Biochemistry, Institute of Integrative Biology, University of Liverpool, Liverpool L69 7ZB, UK; dgfernig@liverpool.ac.uk; 3St. Paul’s Eye Unit, Royal Liverpool University Hospital, Liverpool L7 8XP, UK; 4Department of Clinical Infection, Institute of Infection and Global Health, University of Liverpool, Liverpool L6 97B, UK; cwinstan@liverpool.ac.uk; 5Department of Chemistry, University of Liverpool, Liverpool L69 7ZD, UK; ngberry@liverpool.ac.uk

**Keywords:** *Pseudomonas aeruginosa*, ExoU, type 3 secretion system, inhibitor, non-antibiotic antimicrobials

## Abstract

The opportunistic pathogen *Pseudomonas aeruginosa* employs the type III secretion system (T3SS) and four effector proteins, ExoS, ExoT, ExoU, and ExoY, to disrupt cellular physiology and subvert the host’s innate immune response. Of the effector proteins delivered by the T3SS, ExoU is the most toxic. In *P. aeruginosa* infections, where the ExoU gene is expressed, disease severity is increased with poorer prognoses. This is considered to be due to the rapid and irreversible damage exerted by the phospholipase activity of ExoU, which cannot be halted before conventional antibiotics can successfully eliminate the pathogen. This review will discuss what is currently known about ExoU and explore its potential as a therapeutic target, highlighting some of the small molecule ExoU inhibitors that have been discovered from screening approaches.

## 1. Introduction

*Pseudomonas aeruginosa* is a motile, Gram-negative rod-shaped bacterium that causes a wide range of opportunist infections including corneal, soft tissue, urinary tract, and respiratory tract infections; often with high rates of morbidity and mortality [1,2,3,4,5]. It is a major cause of intensive care unit-acquired pneumonia (ICUAP), as well as a known coloniser of patients with cystic fibrosis and those who are immunocompromised [6,7,8]. Furthermore, multidrug-resistant *P. aeruginosa* is a major threat, as recently highlighted when the World Health Organisation (WHO) listed carbapenem-resistant *P. aeruginosa* as the highest priority for the development of new antibiotics [9]. It is, therefore, imperative that efforts are made to develop novel treatments to target this pathogen.

*P. aeruginosa* produces numerous virulence factors (Table 1) along with a complex regulatory network of intra- and inter-cellular signals allowing it to adapt, thrive, and escape host defences [10]. In particular, the Type-III secretion systems (T3SS) has been identified as a major virulence determinant for poor clinical outcomes in intensive care unit (ICU)-acquired pneumonia (ICUAP), keratitis (infection of the cornea), and otitis externa (infection of the ear canal) [4,5,11,12,13]. The T3SS comprises needle-like, membrane-anchored, multi-component complexes on certain pathogenic bacteria—including *Pseudomonas, Salmonella, Shigella, and Yersina spp.*—that inject effectors from the bacteria’s cytosol directly into the cytoplasm of the eukaryotic host cell [14]. There are four distinct exoenzymes that *P. aeruginosa* employs to project virulence and which are injected into the target host cell via the T3SS: ExoS, ExoT, ExoU, and ExoY. 

*P. aeruginosa* isolates can be broadly categorised into those that carry the gene for ExoS or those that carry the gene for ExoU. Expression of these two exotoxins is almost mutually exclusive and strains of *P. aeruginosa* encoding both or neither are rare [43]. ExoU and ExoS facilitate distinctive mechanisms of bacterial propagation and pathogenesis, which may reflect why they are not cooperatively expressed [43]. While ExoS expression has been associated with endocytic uptake and intracellular survival of bacteria, ExoU expression mediates rapid destruction of the host cell plasma membrane. Endocytic uptake, mediated by ExoS, refers to internalisation of the *P. aeruginosa*, by the host cell, through the invagination of the plasma membrane and the formation of membrane vesicles, in which the bacteria can survive.

ExoS is a bi-functional molecule with an N-terminal GTPase-activating (GAP) domain and a C-terminal ADP-ribosyltransferase domain [44]. It blocks the reactive oxygen species burst in neutrophils by ADP-ribosylation of Ras, thereby preventing the activation of phosphoinositide-3-kinase (PI3K), which is required to stimulate the phagocytic NADPH-oxidase [45]. Injection of ExoS into the target cell also alters the cell cytoskeleton, resulting in cell rounding, inhibition of phagocytosis, which eventually leads to apoptotic cell death [46]. 

ExoU possesses potent phospholipase activity, which causes rapid cell lysis and necroptosis in epithelial cells, macrophages, and neutrophils [29,47,48,49,50]. Furthermore, the depletion of neutrophils leads to an immunosuppressive effect that makes the host more susceptible to secondary infections [50,51]. In comparison to strains expressing ExoS, ExoU-expressing *P. aeruginosa* have been associated with more severe outcomes in keratitis, acute pneumonia, and ICUAP [5,8,43,52].

Given its contribution to clinical severity, therapeutic targeting of ExoU could perhaps attenuate the morbidity and mortality of acute *P. aeruginosa* infections. This review focuses on ExoU; exploring its mechanism of action, as well as the current knowledge about small molecules that have the potential to be developed as inhibitors of ExoU in disease.

## 2. Exotoxin U

ExoU is a 74-kDa (687-amino acid) soluble protein that possesses an N-terminal bacterial chaperone interacting domain followed by a patatin-like phospholipase (PLP) domain and finally C-terminus containing a 4-helic bundle, which is employed for insertion into plasma membranes [29,47,48,53]. It was discovered in vitro and in vivo that ExoU translocation, alone, into mammalian host cells via the T3SS results in rapid cellular necroptosis due to ExoU phospholipase activity directed towards the plasma membrane [47,54,55,56,57]. Although the mechanisms of ExoU activation have yet to be fully explored, it is understood that certain eukaryotic host co-factors directly interact with ExoU and are required for catalytic phospholipase activity to be induced. In-vitro phospholipase activity of recombinant ExoU was only first apparent in the presence of mammalian cell lysate, which indicated that eukaryotic co-factors are required for enzymatic activity [29,48]. Subsequently, ExoU was reported to be activated by the protein superoxide dismutase 1 (SOD1) in vitro [58]. However, it was later discovered that ubiquitinated SOD1, in commercial extracts, was responsible for the activation of ExoU [59]. Ubiquitin polymers, of several linkage types, including linear diubiquitin, have been shown to greatly enhance the ability to bind to and activate ExoU compared to monomeric forms [30]. When ubiquitin and ExoU are co-expressed in *Escherichia coli*, rapid degradation of the bacterial membrane and cell death can be observed, suggesting that ubiquitin is indeed a necessary activating co-factor [30,60]. 

When bound to SpcU, ExoU is inactive, and in the absence of activating eukaryotic co-factors, ExoU expression is not lethal to the *P. aeruginosa*. SpcU is a 137 amino acid, curved, 5-stranded sheet flanked by alpha-helices, which binds to ExoU at the N-terminus [61]. As well as maintaining ExoU in an inactive state, SpcU guides ExoU to the T3SS machinery, for appropriate secretion. SpcU allows ExoU to unfold and pass through the needle of the T3SS [62,63] but does not enter the host cell along with ExoU [14]. After injection, ExoU is activated by binding to ubiquitin before it localises to the plasma membrane where it oligomerises and exerts its phospholipase activity (Figure 1) [48,64]. 

Sequence alignments of ExoU, patatins (*Arabidopsis thaliana* Patatin-like protein 1), mammalian calcium-dependent cytosolic PLA_2_ (cPLA_2_), and calcium-independent PLA_2_ (iPLA_2_) enzymes demonstrate homology in three highly conserved regions; a glycine-rich nucleotide-binding motif between amino acids 111–116, a catalytic serine motif spanning amino acid 140–144, and an active site aspartate residue at amino acid 344–346 [29,48,65,66,67] (Figure 2). Like mammalian PLA_2_ enzymes, ExoU relies on an indispensable catalytic dyad of serine (Ser_142_) and aspartic acid (Asp_344_) [29]. Mutation of either of these residues to an alanine renders the protein inactive in vitro, as well as the abolition of ExoU mediated cytotoxicity in vivo [29].

The crystal structure of ExoU in complex with its cognate chaperone SpcU was solved in 2012 [64,68] (PDB entry numbers 3TU3 and 4AKX). A PLA_2_ (phospholipase) domain between residues 102–471 that contains the active site as well as a flexible region known as the active site ‘cap’ between Pro 320 and Leu 328, which suggests that conformational rearrangement is required for enzymatic activity [64,68]. The structure reveals that the catalytic domain of ExoU is structurally similar to human cPLA_2_ and plant patatin PLA_2_ enzymes [64,68]. The four-helix bundle membrane localisation domain (MLD) of the C-terminus appears to share common structural elements in the targeting of a wide variety of diverse bacterial toxins to the membrane interface. These include clostridial glucosylating toxins, MARTX toxins, and *Pasteurella multocida*-like toxins [69,70].

### 2.1. Secretion of ExoU via the T3SS

The structure of the T3SS comprises a macromolecular complex spanning the bacterium’s internal membrane, periplasmic space, peptidoglycan layer, outer membrane, then the extracellular space and the target cell membrane [71,72]. Upon contact with the target cell, a translocation pore can form across the host cell membrane, allowing the T3SS injectisome to actively secrete the exotoxins from the bacterial cytosol into the host cytosol under the influence of ATPases [14,73,74,75]. The T3SS injectisome has a narrow lumen of approximately 25–40 Å diameter, formed by an assembly of helical PscF proteins [72,76,77]; thus, the exotoxins have to unfold to pass through the channel. The mechanism by which ExoU refolds once inside the host cytoplasm is unknown. 

### 2.2. Control of T3SS Gene Expression

Expression of the T3SS is a controlled response to particular environmental stimuli, such as host cell contact and low levels of extracellular calcium ions [78]. Expression is controlled principally by the interactions of four transcription factors: ExsA, ExsC, ExsD, and ExsE, with transcription factor ExsA considered to be the master regulator [79]. The injectisome, in non-inducing conditions, is expressed at a basal level and is the sensor of inducing signals. In response to host cell contact or low levels of calcium ions, the injectisome converts to a secretion-competent state through an unclear mechanism [79]. In non-inducing circumstances, the transcription factor ExsE remains cytoplasmic in a 1:2 complex with ExsC [80], with ExsA is sequestered by ExsD in a 1:1 complex [81]. However, in inducing conditions, ExsE dissociates from ExsC and translocates through the injectisome [82]. This causes ExsC to bind to ExsD in a 2:2 complex, resulting in the release of ExsA [83], which then targets T3SS promoters to upregulate expression. 

### 2.3. Oligomerisation and Localisation to the Host Cell Wall in the Presence of Phosphatidylinositol 4,5-Bisphosphate (PIP_2_) 

Following translocation into the target cell, ExoU binds to ubiquitin and localises to the plasma membrane by its interaction with phosphatidylinositol 4,5-bisphosphate (PIP_2_) [84,85,86]. PIP_2_ is a biologically important phospholipid in eukaryotic cells, which mostly resides in the inner plasma membrane at relatively low concentrations (~1%) [87]. It is involved in cell signalling pathways, which govern cell adhesion, mobility, cytoskeletal organisation and dynamics, and membrane trafficking [88,89,90,91]. Additionally, it directly binds to focal adhesion molecules, such as talin and vinculin, and other adaptor proteins that have a crucial role in cell–matrix and cell–cell adhesion [90,91].

The C-terminal MLD of ExoU, containing the four-helical bundle, has a high binding affinity for PIP_2_ [68,92]. Models suggest that binding to PIP_2_ causes conformation changes in the structure of ExoU, including the conformational rearrangement of the four-helical bundle of ExoU, allowing it to insert into the lipid membrane [84,93]. PIP_2_ binding has also been demonstrated to promote ExoU multimerisation [93]. SEC-MALS analysis and phospholipase assays indicate that, in vitro, in the presence PIP_2_, ExoU can form multimers that have greatly enhanced catalytic activity, in the presence of ubiquitin, when compared to ExoU and ubiquitin alone [93]. 

Following binding to PIP_2_, ExoU hydrolyses the substrate with a dose-dependent effect on ExoU-mediated cytotoxicity [68,86,92]. A HeLa cell infection model demonstrated that the addition of exogenous PIP_2_ increased ExoU-mediated cytotoxicity [92]. As PIP_2_ co-activates ExoU and enhances in vitro phospholipase activity in the presence of ubiquitin [60,86], it has been postulated that ExoU utilises PIP_2_ to engage the plasma membrane and hydrolyse adjacent substrate phospholipids more efficiently [59,94]. Time-lapse fluorescent image analysis of infected HeLa cells demonstrated that ExoU intoxication correlates with significant intracellular changes leading to necroptosis. Early stages of ExoU intoxication cause disruption of focal adhesions and linkages between integrins, the actin cytoskeleton and the cell membrane, which leads to cell detachment, cytoskeletal collapse, and cell rounding. Subsequently, 3.5 h after infection, membrane blebbing becomes apparent, followed by the loss of plasma membrane integrity, function, and rupture [92]. 

### 2.4. Interactions with Host Cell Signalling Pathways 

It has been suggested that ExoU, as well as inducing cell lysis of the target cell, attenuates particular cellular signalling pathways (Figure 3) [95]. Cuzick et al. demonstrated in a human bronchial cell line that ExoU-producing *P. aeruginosa* could activate the c-Jun NH_2_ terminal kinase (JNK) mitogen-activated protein kinase (MAPK) pathway, resulting in increased production of active AP-1 transcription factor [96]. These findings complement the earlier discovery by McMorran et al., that ExoU-producing *P. aeruginosa* induced the early and transient up-regulation of gene expression in the AP1 transcription factor complex [97]. Cuzick et al., in the same study, demonstrated that catalytically active ExoU, but not the catalytic dead (S142A), significantly increased IL-8 production, suggesting this results from activation of the JNK/MAPK pathway [96]. IL-8 is an important pro-inflammatory chemokine that attracts neutrophils. Effects of neutrophil recruitment could be imported for bacterial clearance; however, ExoU could cause neutrophils to mediate tissue damage and subvert the innate immune response by increasing epithelial cell permeability and, in turn, potentiate invasion of the bacteria [96]. ExoU-positive *P. aeruginosa* has also been shown to trigger the arachidonic acid-dependent inflammatory cascade, inducing a significant release of prostacyclins, PGE_2_ and PGI_2_ (prostacyclin) [98]. 

## 3. Pharmacological Targeting of the Bacterial Phospholipase ExoU

Although the mechanisms of ExoU regulation are not fully understood, evidence for ubiquitin-binding as an activator and recruitment to the plasma membrane and oligomerisation followed by phospholipase A2 activity have been described in ExoU associated pathology [48,64]. These result in multiple dynamic conformational changes that could be targeted by small molecules to attenuate ExoU activity in clinical infections (Figure 1) [84,85,93,99]. 

In 2003, methyl arachidonyl fluorophosphonate (MAFP), an irreversible active site-directed promiscuous phospholipase inhibitor, was shown to protect Chinese Hamster Ovary (CHO) cells from ExoU mediated cell lysis after infection with the PA103 clinical isolate strain of *P. aeruginosa* [48]. The arachidonic acid analogue MAFP is a pan PLA_2_ inhibitor (Figure 4), which possess a phosphate group that covalently binds to the phospholipase catalytic serine residue. Although high concentrations of MAFP (67.5 μM) were required to achieve measurable inhibition of CHO cell lysis, this was the first empirical proof that ExoU catalytic activity could be targeted by a small molecule inhibitor. This also validated the assertion that clinical phospholipase inhibitors, developed to target endogenous human phospholipases across a broad spectrum of diseases [100] could potentially have ExoU inhibiting activity. Such compounds could be repurposed to circumvent the pathological consequences of ExoU expression [101]. 

A subsequent study employed a high-throughput screen to discover compounds that could inhibit T3SS mediated cytotoxicity, with ExoU as the only cytotoxic effector [99]. By maintaining CHO cell viability in the presence of ExoU expressing strains of *P. aeruginosa*, 9H-fluorene-4-carboxamide, the designated pseudolipasin A (Figure 4), was identified as a promising compound lead. Further experimentation confirmed that pseudolipasin A did not inhibit type III secretion or type III injection of ExoU into CHO cells, but rather inhibited ExoU downstream of injection by the T3SS [99]. Relatively high concentrations of pseudolipasin A (5–10 µM) were required to subvert the lytic effects of ExoU after CHO cells were infected with *P. aeruginosa*. This was in agreement with a 7 μM reported IC_50_ value for inhibition for ExoU catalytic activity in vitro, which was determined by a phospholipase assay using recombinant ExoU expressed in *E. coli*. 

In an independent screen, an arylsulphonamide was found to be effective at reducing the cytotoxic effect exerted by induced recombinant ExoU expression in yeast [102]. *Saccharomyces cerevisiae* were transformed with pDH105, which encoded ExoU cDNA. Upon induction with copper, ExoU expression led to *S. cerevisiae* lysis except in the presence of 5 µM of a ‘hit’ arylsulfonamide compound (Figure 4) [102]. Synthesis of a small series of arylsulfonamides, comprising the N-(naphthalen-1-yl)benzenesulfonamide core, validated that this scaffold, when coupled to various commercially available sulfonyl chlorides, yielded compounds with differential abilities to mitigate ExoU mediated cell death [102]. Even though none were as effective as pseudolipasin A, the finding highlighted a novel pharmacological scaffold that could potentially be developed as an inhibitor of ExoU activity. A full biochemical exploration of the inhibitory mechanisms of these compounds remains to be undertaken, and it is not known whether or not these arylsulphonamides inhibit ExoU phospholipase activity in vitro.

Aside from modulating catalytic activity by binding to the ExoU active site, alternate modes of inhibition could occur. For instance, compound binding may inhibit ubiquitin-binding, conformational rearrangement, oligomerization, or phospholipid binding (Figure 1). It also remains to be tested whether or not the arylsulphonamide compounds can effectively protect mammalian cells after *P. aeruginosa* infection in vitro [102]. 

In order to capitalise on pseudolipasin A and arylsulphonamides as potential drug leads, further avenues of structure–activity relationships require exploration. As well as understanding mechanisms and conformations relevant to compound binding, structure-guided drug design could be used to modify pseudolipasin A [99] and the arylsulphonamides [102] to generate selective and more potent ExoU inhibitors. Analogues of the former compound, with a fluorene scaffold, appear to have cytotoxic effects on mammalian cells [102], which could preclude further development of pseudolipasin A as an ExoU inhibitor. Prospectively, the arylsulfonamide potential ExoU inhibitors display limited cytotoxicity in HEK293T cells [102]. Crystal structures of ExoU in complex with ubiquitin, phospholipid and/or compounds would be valuable for informing structure-guided design of such compounds. ExoU appears to be a dynamic protein, and the study of such conformations associated with cofactor and ligand binding might be explored by cryogenic electron microscopy (cryo EM). As ExoU is catalytically inactive when bound to its cognate chaperone SpcU [64], the already available co-crystal structures of ExoU and SpcU (PDB entries 3TU3 and 4AKX) [64,68] could potentially serve as a foundation for structure-based design of inhibitors or peptides that mimic the interface between the N-terminal domain of ExoU and SpcU. Perhaps the structures of ExoU and SpcU could be explored by in sillico-based screening approaches in order to discover novel potential ExoU ligands. 

### 3.1. Allosteric Inhibitors 

In order to find novel ExoU inhibitors, multiple screening strategies could be explored to evaluate compounds’ ability to inhibit phospholipase A2 ExoU activity, translocation through the T3SS, and interaction with host co-factors. Allosteric inhibitors of ExoU may offer distinct efficacious and selective advantages over small molecules occupying the phospholipase substrate-binding pocket. Molecules that bind to conserved regions of the active site, encompassing the Serine-Aspartic acid catalytic dyad, may have limited selectivity and produced unwanted ‘off-target’ cytotoxic effects. Indeed, pseudolipasin A does not inhibit any of human group IID, IIE, V, X, or XII PLA_2_ phospholipases tested [99]. Allosteric inhibitors exert their effects either by indirectly targeting catalytic activity or by modulating non-catalytic function, such as interactions with activating co-factors. They generally achieve better target precision through binding to specific residues, outside of the conserved active site. Such approaches have become popular for pharmacological targeting of disease-associated protein kinases [103]. Exploiting the ubiquitin-binding site of ExoU, which has been modelled in silico with the use of biophysical analysis and available crystallographic evidence [104], could serve as a point of allosteric inhibition that is selective and potent to ExoU activity inhibition. Such inhibitors could offer the advantage of being non-substrate competitive and, therefore, potentially, achieve superior potency. To this end, screening approaches to interrogate ExoU interaction with ubiquitin and PIP_2_ in the presence of prospective compounds would need to be developed. 

An example of allosteric targeting of phospholipases is compound VU0364739. This is a phospholipase D2 (PLD2)-selective inhibitor, which occupies an allosteric site and blocks PIP_2_ binding [105]. The selectivity of this compound was evidenced by a starkly increased IC_50_ value for inhibition of the closely related PLD1 isoform of 1500 nM compared to the IC_50_ value for PLD2 of 20 nM. VU0364739 is structurally similar to previously reported piperidine benzimidazolone-based allosteric inhibitors of AKT [106]. The PIP_2_-binding site forms a hydrophobic pocket in the pleckstrin homology (PH) domain of PLD2, and the mode of inhibition is proposed to be akin to that facilitated by allosteric AKT compounds [105,107]. Perhaps a similar allosteric mechanism of inhibition of ExoU could be discovered, whereby certain classes of compound occupy a site within the C-terminal domain of ExoU in order to mitigate interaction with PIP_2_. 

### 3.2. Ligand Repurposing Approaches

The activities of phospholipase A2 enzymes is a contributing factor to various inflammatory pathological conditions, including arthritis, cardiovascular, and autoimmune diseases [100]. Modulation of their catalytic activity has become of major interest to pharmaceutical companies in recent years for the development of novel therapeutics [100]. De novo drug discovery, usually performed in the pharmaceutical industry, is an expensive and long process. Drug repurposing approaches have been used across a broad spectrum of diseases [108], where many “repurposed” drugs, such as antibiotics, antivirals, and kinase inhibitors, have had clinical success in treatment areas beyond their originally intended use. In a recent study, certain kinase inhibitors, including an inhibitor of PDK1, were able to function as an adjuvant in colistin-resistant bacteria, potentiating the antimicrobial effect in Gram-negative and Gram-positive bacteria [109]. This demonstrates that cross-reactivity of inhibitors originally intended to target eukaryotic proteins may be extended to target prokaryotic virulence effectors. Repurposing of clinical eukaryotic phospholipase inhibitors, to instead target the phospholipase A2 activity of ExoU, may have the advantage of entering the clinic more quickly, as they have already been tested for safety in human beings [101]. 

There may yet exist phospholipase A2 inhibitors that are cross-reactive with ExoU, which could be utilised when ExoU expression by *P. aeruginosa* is a driver of disease progression. Compounds exhibiting such cross-reactivity with ExoU could potentially serve as a foundation for the development of more selective ExoU inhibitors. Medicinal chemistry approaches, including in silico modelling, organic synthesis, and the study of structure-activity relationships may provide new chemicals as potential novel anti-ExoU pharmaceutical agents. 

## 4. Conclusions 

The type III secretion system of *P. aeruginosa* is used to inject the toxins ExoS, ExoT, ExoU, and ExoY into the cytosol of target eukaryotic cells during infection. ExoU is characterised as the major virulence factor responsible for acute epithelial injury in numerous diseases, including corneal, soft tissue, urinary tract, and respiratory tract infections [1,2,3,4,5]. Given the accessibility of the eye to the study of infections and topical treatment, as well as the predominance of ExoU-positive phenotypes in severe infection, it might be advantageous to use the cornea as a model to study the effect of drugs which target ExoU. In therapeutic contexts, as well as mitigating acute cellular damage, targeting ExoU may also have the advantage of a decreased risk for selecting resistance, as inhibition of ExoU mediated virulence may synergise with and allow the host immune system to respond better to the threat [96]. By subverting cellular signalling caused by ExoU, as well as ExoU phospholipase activity, neutrophils, of the innate immune system may be better equipped to respond to *P. aeruginosa* virulence [96]. Indeed, pharmacological targeting of ExoU may be compatible with antibiotic usage, whereby inhibitors of ExoU serve as an adjuvant therapy. In this way, ExoU inhibitors could mitigate the acute cytotoxic effects whilst conventional antibiotics eliminate the *P. aeruginosa*.

As well as new strategies to discover more compound leads, further structural evidence is required to direct the successful structure-guided design of current ExoU inhibitors. The development of biochemical assays should serve to discern the potential mechanisms of inhibition, be it by occupying the active site or preventing an interaction with an important activating co-factor.

## Figures and Tables

**Figure 1 microorganisms-07-00707-f001:**
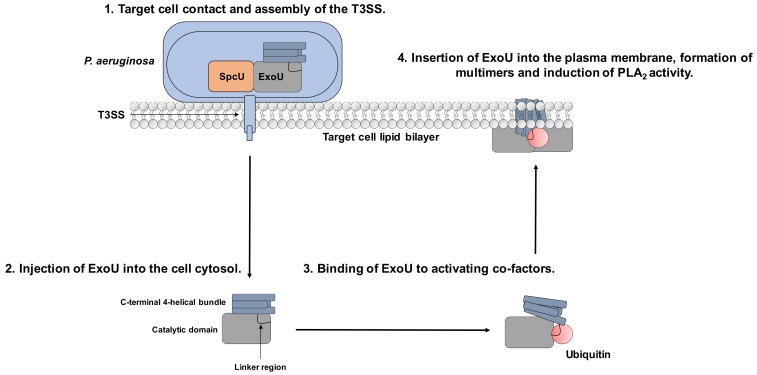
Mechanism of ExoU toxicity and potential for targeting with small molecules. 1. *P. aeruginosa* binds to the target cell, which stimulates T3SS assembly. 2. ExoU is injected to the host cell via the T3SS. 3. Once in the cellular cytoplasm, it interacts with the eukaryotic host co-factor ubiquitin. 4. ExoU localises to the plasma membrane and oligomerises to stimulate full catalytic activity, leading to cellular lysis. The linker region links the catalytic domain to the 4-helical membrane localisation domain (MLD) and is the proposed binding site for ubiquitin. The 4-helical MLD domain is in the C-terminus of the protein and is responsible for interaction with PIP_2_ and insertion into the host cell plasma membrane. Compounds that could potentially subvert ExoU mediated cytotoxicity could: prevent ExoU secretion from *P. aeruginosa*, bind to the substrate-binding site in the catalytic domain, prevent ubiquitin-binding, or inhibit multimerisation and association with PIP2.

**Figure 2 microorganisms-07-00707-f002:**
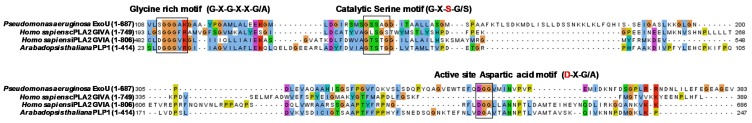
Amino acid alignment of *P. aeruginosa* ExoU, human cPLA_2_ GIVA, human iPLA_2_ GVIA and *Arabidopsis thaliana* PLP1. The full-length sequences of ExoU (uniprot O34208), human cPLA_2_ α GIVA (uniprot P47712), human iPLA_2_ GVIA (uniprot O60733), and *Arabidopsis thaliana* Patatin-like protein 1 (PLP1) (uniprot O23179) were aligned using the TcoffeeWS multiple sequence alignment tool and visualised in Jal view. The glycine-rich, catalytic serine and catalytic aspartic acid motifs are shown in black boxes.

**Figure 3 microorganisms-07-00707-f003:**
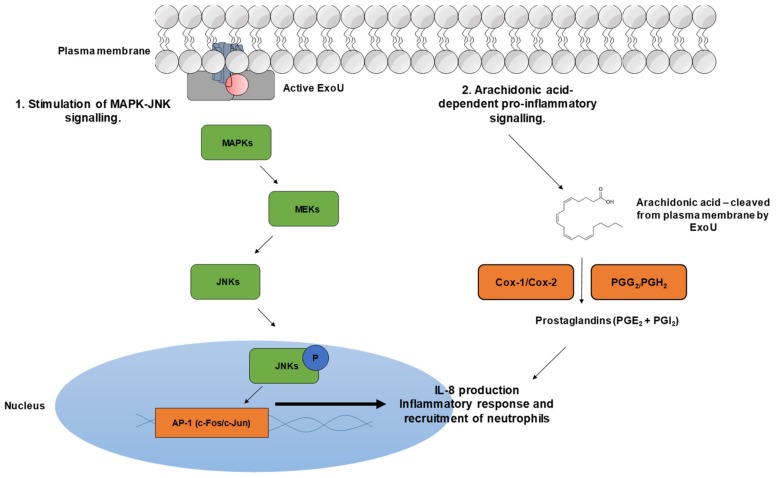
ExoU mediated activation of host cell signalling pathways. 1. The injected active ExoU stimulates the mitogen-activated protein kinase (MAPK) signalling cascade through an unknown mechanism. MEKs phosphorylate c-Jun NH2 terminal kinase (JNKs), which translocate to the nucleus and activate c-Fos and c-Jun transcription factors to stimulate the inflammatory response. 2. Active ExoU cleaves membrane phospholipids at the sn2 position to yield arachidonic acid. Arachidonic acid is converted to prostaglandins PGE_2_ and PGI_2_ by cyclooxygenases Cox-1 and Cox-2 and endoperoxides PGG_2_ and PGH_2_.

**Figure 4 microorganisms-07-00707-f004:**
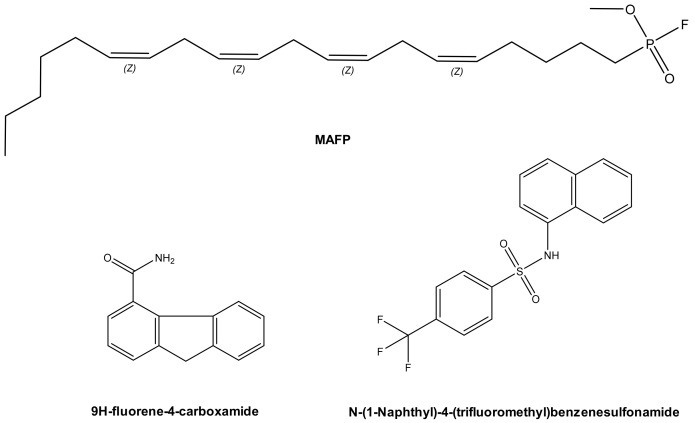
Prospective ExoU small molecule inhibitors. MAFP (Methoxy arachidonyl fluorophosphonate) is a non-specific PLA_2_ inhibitor, which is an analogue of arachidonic acid. Chemical structures of compounds previously distinguished to inhibit ExoU from two independent screens are also illustrated.

**Table 1 microorganisms-07-00707-t001:** Important virulence effectors of *Pseudomonas aeruginosa* in mammalian infection.

Virulence Factor	Category	Activity	Function	Ref.
Alginate	Extracellular polysaccharide	Biofilm formation	Contributes to biofilm formation and reduces susceptibility to antibiotics.	[15,16]
Alkaline protease (ArpA)	Exoenzyme	Zinc-dependent metalloprotease	Degrades host immune complements C1q, C2, and C3 and cytokines interferon (IFN)-γ and tumour necrosis factor (TNF)-α.	[17,18,19]
Cystic fibrosis transmembrane conductance regulator (CFTR) inhibitory factor (Cif)	Phenazine	Epoxide hydrolase	Promotes sustained inflammation by hydrolysing the paracrine signal 14,15-epoxyeicosatrienoic acid that stimulates neutrophils to produce the pro-resolving lipid mediator 15-epi lipoxin A4. Cif increases the ubiquitination and lysosomal degradation of some ATP-binding cassette transporters (ABC) including CFTR, P-glycoprotein, and TAP1.	[20,21,22,23]
ExoA	Exoenzyme	Catalytic ADP-ribosylation of elongation factor 2	Inhibits protein synthesis and induces apoptosis in the host cell.	[24]
ExoS	Exoenzyme	Bifunctional toxin with Rho GTPase-activating protein (RhoGAP) activity and ADP-ribosyltransferase (ADPRT) activity	It blocks the reactive oxygen species burst in neutrophils by ADP-ribosylation of Ras, thereby preventing the activation of phosphoinositide-3-kinase (PI3K), which is required to stimulate the phagocytic NADPH-oxidase.	[25,26]
ExoT	Exoenzyme	Bifunctional toxin with RhoGAP activity and ADPRT activity	It impairs the production of reactive oxygen species burst in neutrophils and promotes the apoptosis of host cells by transforming host protein Crk by ADP-ribosylation into a cytotoxin and by activation of the intrinsic mitochondrial apoptotic pathway.	[27,28]
ExoU	Exoenzyme	Phospholipase A2	It becomes activated by interaction with ubiquitin or ubiquitinylated proteins in the cytosol of the host cell before localising to the cell membrane to catalyse fatty acids from a broad range of phospholipids and lysophospholipids.	[29,30]
ExoY	Exoenzyme	Secreted adenyl cyclase	Increases concentration of intracellular cAMP in host cells through disruption of the actin cytoskeleton and increased endothelial permeability.	[28]
Flagella	Organelle	Motility and adherence to surfaces	Elicits strong NFκB-mediated inflammatory response via signalling through toll-like receptor (TLR) 5 and a caspase-1-mediated response through Nod-like receptor, Ipaf. Provides bacterium with swimming motility in liquid.	[31]
LasA	Exoenzyme	Metallopeptidase, also known as staphylolysin	LasA acts with restricted specificity, predominantly at glycine-glycine peptide bonds, but also increases the elastinolytic activity of LasB.	[32]
LasB	Exoenzyme	Zinc-metalloprotease	Causes elastin degradation.	[17,33]
PlcH	Exoenzyme	Haemolytic phospholipase C	Releases phosphate esters from sphingomyelin and phosphatidylcholine.	[34]
PlcN	Exoenzyme	Non-haemolytic phospholipase C	Releases phosphate esters from phosphatidylserine and phosphatidylcholine.	[35]
PldA and PldB	Exoenzyme	Phospholipase D	Facilitates intracellular invasion of host eukaryotic cells by activation of the PI3K/ Akt pathway.	[36,37]
PrpL	Exoenzyme	Class IV protease, lysine endoproteinase	Inactivates a range of host defences including fibrinogen, plasminogen, immunoglobulin G, and complement proteins C1q and C3.	[38,39]
Pyocyanin	Phenazine	Redox-active zwitterion	Inhibits host cell respiration, ciliary function, and epidermal growth; disrupts calcium homeostasis and induces apoptosis in neutrophils.	[40]
Rhamnolipids	Surfactant	Biosurfactants	Participates in the maintenance of uninhabited channels surrounding biofilm communities, which serve to provide nutrients and oxygen to the colonies of bacteria. Biofilms can form on implants and on dead or living tissue. They are inherently difficult to eradicate with antibiotics due to the inability of antibiotic molecules to penetrate the extracellular matrix.	[41]
TplE	Exoenzyme	Phospholipase A1	Disrupts the endoplasmic reticulum and thereby promotes autophagy by the activation of the unfolded protein response.	[42]

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
