# Peer review of "Pseudomonas aeruginosa* Toxin ExoU as a Therapeutic Target in the Treatment of Bacterial Infections"

_microorganisms, 2019, doi:10.3390/microorganisms7120707_

Round 1
Reviewer 1 Report
Dear Editor and Authors,
The authors have written a review of the Pseudomonas aeruginosa toxin ExoU. They review the expression, secretion, binding and enzymatic activity of ExoU. The authors proposed that small molecule therapies might be developed to interfere with secretion of ExoU, its enzymatic activity, its binding to ubiquitin, or its target cell membrane localization/oligomerization. They reviewed some of the small molecules that have been used to inhibit its enzymatic activity. None of the molecules reviewed were useful therapeutic agents. The authors reviewed the possibility that some small molecule allosteric modulators of ExoU may have some therapeutic utility.
The authors use British spellings throughout the manuscript (utilises, localises, oligomerises, etc.). The Journal does not specify which form of the English language should be used. I leave that issue to the editor. There are more important errors to correct.
Each of the references is incomplete, formatted incorrectly or both. There are at least three different examples of a reference being duplicated in the reference list. These errors need to be corrected.
Figure 1 is confusing. The scale differences of the very components are counterfactual. The lipid bilayer is very large. By comparison, the bacteria are quite small. The enzymes are larger than the lipid bilayer, but not by much. Ideally, the authors should redraw the figure. They should increase the text font size within the figure and reduce the amount of text. The caption can be used to convey any extra information. It would help if the authors indicated that the large grey spheres attached to the hooked squiggles are part of the target cell’s lipid bilayer. The rectangles with flagella should be labeled P. aeruginosa – or + T3SS. Differentiating the 4 helical bundles, b-sheet and linker region by color would visually reinforce the different regions.
Figure 2 is prepared in a way that will make it very difficult to read. This reviewer is not sure that anyone will be able to read the figure due to the small font size. Since the authors are focusing on only two regions 108-200 and 305-383, they should break each of those regions into two 50 amino acid portions and stack these. The figure will be taller, but it will allow the authors to increase the text font size, so it will be visible. The catalytic serine motif and the active site Aspartic acid motif will remain intact in their portions of the figure.
Figure 3 contains errors that need to be corrected. The structure of MAFP is incorrect. In line drawings of organic molecules an edge indicates a covalent bond, a double edge indicates a double bond, a vertex indicates the presence of a carbon atom, unless a heteroatom is indicated; the presence of the requisite number of hydrogens is implied. For example, ethane (CH3-CH3) is represented by a simple edge with two vertices (-), corresponding to single covalent bond between two carbon atoms, each bound, by implication, to three hydrogens. The authors added extra edges to each carbon vertex. They neglected to place a H on the vertex of those extra edges to explicitly indicate a hydrogen. As it is drawn, MAFP contains extra edges and vertices which, by convention, indicate the presence of methyl (CH3) groups and not the intended hydrogens.
Other comments and suggestions are listed below.
Line 43 propagate virulence perhaps project virulence, impose virulence
Line 94 P. aeruginosa should be P. aeruginosa
Line 94 binds to the host cell should be binds to the target cell, since the P. aeruginosa is not going to reside inside the eukaryotic cell.
Lines 113-114 read: “The crystal structure of ExoU in complex with its cognate chaperone SpcU was solved in 2003 (48).”
Reference 48 does not report a crystal structure; it describes a mechanism of action. The correct references are the 2012 references, 71 and 72. The authors should include the pdb entry numbers (3TU3 and 4AKX), so interested readers can visualize the structures.
Lines 123-130. The authors choose to use the term “host cell”, which implies that Pseudomonas aeruginosa will reside inside of it. The term “target cell” seems to be more appropriate, since Pseudomonas aeruginosa T3SS is targeting a cell for injection of toxins that will soon kill that cell.
Lines 145-146 read as, “Following translocation into the host cell, ExoU binds to ubiquitin and localises to the plasma membrane by its interaction with phosphatidylinositol 4,5-bisphosphate (PIP2) (48,62,88–92). Ref 48 describes mechanism of ExoU. Ref 88 describes chaperon for secretion of ExoU. Ref 92 describes cytoskeletal collapse. Are these three references appropriate for the statement?
Lines 172-177 et al. is an abbreviation for Latin phrase et alia, it should be et al. and not et al .
References 28 and 31 are identical.
References 29 and 48 are identical.
References 92 and 98 are identical.
Lines 201-202: This sentence “Such compounds could undergo repurposing and circumvent pathology where ExoU expression is a driver of disease progression (107).” Is unnecessarily confusing. Perhaps, “Such compounds could be repurposed to circumvent pathological consequences of ExoU expression (107).”
Lines 205-206 The compound should be 9H-fluorene-4-carboxamide.
Line 206: ”carboxamide, a designated pseudolipasin A (Figure 3)” the “a” should be deleted.
Lines 203-212. pseudolipasin A should not be capitalized.
Line 236: “further avenues of structural biology require exploration” should be “further avenues of structure activity require exploration”
Figure 3.
9h -fluorene-4-carboxylic acid amide is wrong, it should be: 9H-fluorene-4-carboxylic acid amide. To be consistent, the authors should decide on using 9H-fluorene-4-carboxylic acid amide or 9H-fluorene-4-carboxamide throughout the manuscript.
Lines 205-206 The compound should be 9H-fluorene-4-carboxamide.
The references are improperly formatted.
For example, reference 1 Newman JW, Floyd R V., Fothergill JL. The contribution of Pseudomonas aeruginosa virulence factors and host factors in the establishment of urinary tract infections. FEMS Microbiol Lett. 2017;364(15):1–12.
Should be formatted as listed:
Newman, J.W.; Floyd, R.V.; Fothergill, J.L. The contribution of Pseudomonas aeruginosa virulence factors and host factors in the establishment of urinary tract infections. FEMS Microbiol. Lett. 2017, 364:1–12.
Some of the references are incomplete.
Some of the references include the Journal’s publisher.
The authors need to revise the references, so that they are all properly formatted for this Journal and complete.
The reviewer has listed a few examples of the types of corrections
Reference 16 is a chapter form a 2012 book, it needs to be completely referenced as such.
Reference17 Why include the publisher of the journal?
References 18-23, 25, 28 are incomplete, without volume and page numbers.
Reference 24 has a garbled Journal name.
Reference 27 Is an improperly formatted book chapter.
Reference 29 has partial page numbers (2959–69). The should be full page numbers (2959–2969)
Reference 31 Reference lists the Journal as PLoS One. Public Library of Science. It should be PLoS One.
References 34-38 also incomplete, same as 18-23.
Author Response
Response to Reviewer 1
Line 43 propagate virulence perhaps project virulence, impose virulence.
Authors agree "project virulence" is more suitable.
Line 94 P. aeruginosa should be P. aeruginosa.
Amendment made.
Line 94 binds to the host cell should be binds to the target cell, since the P. aeruginosa is not going to reside inside the eukaryotic cell.
Agreed and changed.
Lines 113-114 read: “The crystal structure of ExoU in complex with its cognate chaperone SpcU was solved in 2003 (48).”
Mistake has been rectified.
Lines 123-130. The authors choose to use the term “host cell”, which implies that Pseudomonas aeruginosa will reside inside of it. The term “target cell” seems to be more appropriate, since Pseudomonas aeruginosa T3SS is targeting a cell for injection of toxins that will soon kill that cell.
Instances of “host cell” have been changed to target cell for this section.
Lines 145-146 read as, “Following translocation into the host cell, ExoU binds to ubiquitin and localises to the plasma membrane by its interaction with phosphatidylinositol 4,5-bisphosphate (PIP2) (48,62,88–92). Ref 48 describes mechanism of ExoU. Ref 88 describes chaperon for secretion of ExoU. Ref 92 describes cytoskeletal collapse. Are these three references appropriate for the statement?
More appropriate references have been cited.
Lines 172-177 et al. is an abbreviation for Latin phrase et alia, it should be et al. and not et al .
Changed.
References 28 and 31 are identical.
References 29 and 48 are identical.
References 92 and 98 are identical.
Duplicate references have been removed and bibliography has been properly formatted.
Lines 201-202: This sentence “Such compounds could undergo repurposing and circumvent pathology where ExoU expression is a driver of disease progression (107).” Is unnecessarily confusing. Perhaps, “Such compounds could be repurposed to circumvent pathological consequences of ExoU expression (107).”
Agreed and changed.
Lines 205-206 The compound should be 9H-fluorene-4-carboxamide.
Corrected.
Line 206: ”carboxamide, a designated pseudolipasin A (Figure 3)” the “a” should be deleted.
Corrected.
Lines 203-212. pseudolipasin A should not be capitalized
Corrected.
Line 236: “further avenues of structural biology require exploration” should be “further avenues of structure activity require exploration”
Agreed and changed.
Figure 3.
9h -fluorene-4-carboxylic acid amide is wrong, it should be: 9H-fluorene-4-carboxylic acid amide. To be consistent, the authors should decide on using 9H-fluorene-4-carboxylic acid amide or 9H-fluorene-4-carboxamide throughout the manuscript.
Lines 205-206 The compound should be 9H-fluorene-4-carboxamide.
Agreed and changed.
The references are improperly formatted.
For example, reference 1 Newman JW, Floyd R V., Fothergill JL. The contribution of Pseudomonas aeruginosa virulence factors and host factors in the establishment of urinary tract infections. FEMS Microbiol Lett. 2017;364(15):1–12.
Should be formatted as listed:
Newman, J.W.; Floyd, R.V.; Fothergill, J.L. The contribution of Pseudomonas aeruginosa virulence factors and host factors in the establishment of urinary tract infections. FEMS Microbiol. Lett. 2017, 364:1–12.
Some of the references are incomplete.
Some of the references include the Journal’s publisher.
The authors need to revise the references, so that they are all properly formatted for this Journal and complete.
The reviewer has listed a few examples of the types of corrections
Reference 16 is a chapter form a 2012 book, it needs to be completely referenced as such.
Reference17 Why include the publisher of the journal?
References 18-23, 25, 28 are incomplete, without volume and page numbers.
Reference 24 has a garbled Journal name.
Reference 27 Is an improperly formatted book chapter.
Reference 29 has partial page numbers (2959–69). The should be full page numbers (2959–2969)
Reference 31 Reference lists the Journal as PLoS One. Public Library of Science. It should be PLoS One.
References 34-38 also incomplete, same as 18-23.
The bibliography has been properly formatted properly with the referencing style conducive to the journal.

Reviewer 2 Report
This is a very nice and comprehensive review about the role of ExoU in P. aeruginosa pathogenesis and potential ways to target this enzyme in therapeutic settings. The review is well structured, clear and laid out in a logical order. I have only very minor suggestions for further improvement.
abstract: a statement is made about rates of resistance evolution. I did not see this explained in the main text, and it remains unclear what resistance was against, and how the rates were measured. Presumably the authors are referring to a study that used fluctuation tests, but I would like this to be confirmed, as the term "rate" is often misused. Figure 1 can be significantly improved. Lots of stuff going on in this figure, but it's not intuitively laid out... perhaps redraw figure with different elements closer to scale, and where needed panels where certain processes are zoomed in. I would expect to see ExoU and SpcU in the bacterium ; I would expect a clearer picture of the T3SS and movement of ExoU across the membrane, etc. The authors can take advantage of existing x-ray structures of ExoU and cocrystal structures of ExoU and SpcU and incorporate those instead of the ppt schematics.
section 2.2 seems out of place and can be dropped entirely. Not sure why information about T3SS is relevant to the key message of this review.
Section 2.4 needs a figure to illustrate the pathways that are discussed. Many will be familiar with those pathways, but as many will not.
Section 3: I was surprised that SpcU is not mentioned, as this protein apparently inhibits ExoU and could serve as a model for structure-based design of inhibitors.
In general there is a lot of emphasis on the structural biology of the virulence factors and their targets, but no structures are shown. I would recommend the authors to fix that, as it will help the reader to grasp the structure-function relationship and structural homologies that are suggested to be important.
Minor comments:
L28 - i personally would prefer describing the morphology as "rod-shaped" to avoid confusion with the genus Bacillus.
L39 - please add in brackets a short description of these pathological states in plain english.
Very nice table, but unclear what PlcH and PlcN do and what the consequences are (text provided is not unintelligible to me and does not explain why the described activity matters).
L50 - define "endocytic"
Fig. 1 - "recruteed". ?
Author Response
Response to reviewer 2
abstract: a statement is made about rates of resistance evolution. I did not see this explained in the main text, and it remains unclear what resistance was against, and how the rates were measured. Presumably the authors are referring to a study that used fluctuation tests, but I would like this to be confirmed, as the term "rate" is often misused. Figure 1 can be significantly improved. Lots of stuff going on in this figure, but it's not intuitively laid out... perhaps redraw figure with different elements closer to scale, and where needed panels where certain processes are zoomed in. I would expect to see ExoU and SpcU in the bacterium ; I would expect a clearer picture of the T3SS and movement of ExoU across the membrane, etc. The authors can take advantage of existing x-ray structures of ExoU and cocrystal structures of ExoU and SpcU and incorporate those instead of the ppt schematics.
The rates of resistance and correlation with ExoU status in P. aeruginosa is currently unpublished data. Therefore we have decided to omit this statement form the abstract.
-Figure 1 has been redrawn with suggestions considered and in particular relating to scaling, though we felt that only showing part of the bacterial cell might be confusing for some readers, so this is not to scale. We do not believe it is necessary to employ the crystal structures here, as there are no currently structures of the active form(s) of ExoU, though there is considerable biophysical and biochemical evidence for substantial conformational change between the SpcU complexed ExoU and the lipid and ubiquitin bound enzyme. Therefore, it is appropriate to use a schematic, since we are drawing on indirect structural evidence.
Section 2.2 seems out of place and can be dropped entirely. Not sure why information about T3SS is relevant to the key message of this review.
In the legend to Fig. 1 we allude to the possibility that drugs that interfere with ExoU function may prevent secretion. It seems useful in this respect to have a short section on the T3SS, so that readers can use this as a springboard should they wish to pursue this aspect.
Section 2.4 needs a figure to illustrate the pathways that are discussed. Many will be familiar with those pathways, but as many will not.
We agree that schematics of these pathways would be useful for those who wish to read further in depth and visualise them. Therefore, we have produced an additional figure to illustrate the potential pathways that ExoU is involved in. We have also cited a recent review on the pathways to provide a link for the reader to the large body of literature available.
Section 3: I was surprised that SpcU is not mentioned, as this protein apparently inhibits ExoU and could serve as a model for structure-based design of inhibitors.
Very interesting comment. We have briefly included how the co-crystal structures of ExoU and SpcU may be useful for structure guided design of ExoU N-terminal binding domain inhibitors/peptides.
In general there is a lot of emphasis on the structural biology of the virulence factors and their targets, but no structures are shown. I would recommend the authors to fix that, as it will help the reader to grasp the structure-function relationship and structural homologies that are suggested to be important.
We have indeed focussed on the structural side. We have also avoided a detailed structural analysis using figures generated from the PDB files, because:
(i) The only structures of ExoU are of it bound to SpcU.
(ii) All remaining strcuctural evidence is biochemical, biophysical and/or generated by modelling.
(iii) There is no rigorous molecular dynamics analysis of ExoU structure, which might enable an approximation of the enzyme’s structure in the mammalian cell.
Therefore, we felt that given the limitations of our current structural knowledge providing the PDB accessions and references to the biophysical analyses is sufficient. A static view of the SpcU bound ExoU does not add much; it is better to invite the reader to view the original papers on the structure and biophysical analyses.
Minor comments:
L28 - i personally would prefer describing the morphology as "rod-shaped" to avoid confusion with the genus Bacillus.
We have corrected this
L39 - please add in brackets a short description of these pathological states in plain English.
Agreed, brackets with descriptions have been included.
Very nice table, but unclear what PlcH and PlcN do and what the consequences are (text provided is not unintelligible to me and does not explain why the described activity matters).
These potentially enable acquisition of phosphate under low phosphate conditions and the generation of immunomodulatory signals. The provided reference enables the reader to explore the limited literature on these enzymes.
L50 - define "endocytic"-
A statement about endocytosis has been added.
Fig. 1 - "recruteed". ?- This mistake has been rectified.

Round 2
Reviewer 1 Report
Dear Authors and Editor,
The authors have made the requested changes. The authors removed the methyl groups, but the structure of MAFP is still incorrect. The structure of MAFP (between lines 245-246) contains trans double bonds (hydrogens on adjacent carbons of the double bond are on the opposite side of the double bond). The four double bonds of arachidonic acid are cis (hydrogens on adjacent carbons of the double bond are on the same side of the double bond). The authors will have to accurately redraw the structure.
The authors should include the link to the Ref. 9 WHO report (https://www.who.int/medicines/publications/WHO-PPL-Short_Summary_25Feb-ET_NM_WHO.pdf?ua=1)
More than half of the references are incomplete. This includes references 12, 15, 16, 18-23, 25, 28, 33-37, 39, 42, 43, 45, 46, 50, 51, 54-57, 62-67, 69-76, 79-84, 87-91, 95, 96, 99-101, 103, and 105-109. These references should be completed by the authors and not left to be done by the authors and not left to the copy editors.
Author Response
Dear reviewer
MAFP has been redrawn properly.
The references are now complete. For reference 108 (Schein, C.H. Repurposing approved drugs on the pathway to novel therapies) the DOI has been specified as this article does not yet appear in any of the Medicinal research review volumes.
On behalf of the all the authors, we express or sincerest thanks for your thorough evaluation of the manuscript.
Reviewer 2 Report
The authors addressed all concerns in a satisfactory manner
Author Response
Dear reviewer
On behalf of the all the authors, we express or sincerest thanks for your thorough evaluation of the manuscript and the insightful comments provided.
Round 3
Reviewer 1 Report
Dear Authors and Editor,
The authors have made the requested changes. The structure of MAFP is correct. The manuscript should be published.